# The Effect of Non-Personalised Tips on the Continued Use of Self-Monitoring mHealth Applications

**DOI:** 10.3390/brainsci10120924

**Published:** 2020-11-30

**Authors:** Vishnu Unnikrishnan, Miro Schleicher, Yash Shah, Noor Jamaludeen, Ruediger Pryss, Johannes Schobel, Robin Kraft, Winfried Schlee, Myra Spiliopoulou

**Affiliations:** 1Institute of Technical and Business Information Systems, Otto-von-Guericke-University Magdeburg, 39106 Magdeburg, Germany; miro.schleicher@ovgu.de (M.S.); yash.shah@ovgu.de (Y.S.); noor.jamaludeen@ovgu.de (N.J.); myra@ovgu.de (M.S.); 2Institute of Clinical Epidemiology and Biometry, University of Würzburg, 97070 Würzburg, Germany; ruediger.pryss@uni-wuerzburg.de; 3Institute of Databases and Information Systems, Ulm University, 89081 Ulm, Germany; johannes.schobel@uni-ulm.de (J.S.); robin.kraft@uni-ulm.de (R.K.); 4Department of Psychiatry and Psychotherapy, Regensburg University, 93053 Regensburg, Germany; winfried.schlee@tinnitusresearch.org

**Keywords:** tinnitus, ecological momentary assessments, physician feedback, mHealth, self-monitoring, non-personalised tips

## Abstract

Chronic tinnitus, the perception of a phantom sound in the absence of corresponding stimulus, is a condition known to affect patients’ quality of life. Recent advances in mHealth have enabled patients to maintain a ‘disease journal’ of ecologically-valid momentary assessments, improving patients’ own awareness of their disease while also providing clinicians valuable data for research. In this study, we investigate the effect of non-personalised tips on patients’ perception of tinnitus, and on their continued use of the application. The data collected from the study involved three groups of patients that used the app for 16 weeks. Groups A & Y were exposed to feedback from the start of the study, while group B only received tips for the second half of the study. Groups A and Y were run by different supervisors and also differed in the number of hospital visits during the study. Users of Group A and B underwent assessment at baseline, mid-study, post-study and follow-up, while users of group Y were only assessed at baseline and post-study. It is seen that the users in group B use the app for longer, and also more often during the day. The answers of the users to the Ecological Momentary Assessments are seen to form clusters where the degree to which the tinnitus distress depends on tinnitus loudness varies. Additionally, cluster-level models were able to predict new unseen data with better accuracy than a single global model. This strengthens the argument that the discovered clusters really do reflect underlying patterns in disease expression.

## 1. Introduction

“Tinnitus”, derived from “tinnire”, Latin for *to ring*, is a neuropsychiatric disorder where the patients report the perception of a ‘phantom sound’. Although the name suggests a ringing sound, the symptoms vary, with patients describing their tinnitus as a ringing, rushing, roaring, hissing, or clicking noise. The tinnitus may or may not be pulsatile, and its onset can be abrupt or gradual. The symptomatic presentation of tinnitus is further complicated by the sound being perceived preferentially in one or both ears, as being inside the head, or externally located. Adding further heterogeneity to the tinnitus experience is the fact that the volume can vary over time. While stress is known to exacerbate tinnitus, there is a general consensus that further basic research is necessary to understand the disease better and to improve treatment [1].

Chronic diseases like tinnitus stand to gain greatly from emerging trends in mobile health (or ‘mHealth’). Increased interest in mobile health sensing was primarily driven by research interest in measuring a more ’organic’ presentation of the disease outside of a hospital.mHealth technologies can also help the patient update their doctor on their current symptoms, and collect feedback from the doctor on symptom management without the need to visit a clinic. The benefit of access to the doctor directly from mobile devices may therefore not only improve patient-doctor communication but may also improve access to healthcare for patients with movement constraints or poor access to healthcare. Patient involvement in tracking their own disease state may also improve self-monitoring, awareness, and consequently well-being while providing the physician with a more accurate picture of the patient’s symptoms, namely mitigating the effects of confounders like *recall bias* while maintaining *ecological validity* [2].

The idea of tracking patients’ well-being has been explored through the use of mHealth apps that collect “Ecological Momentary Assessments” (EMAs). EMAs are preferred as the observations are on a patient’s current state, and in the patient’s current environment, which is particularly important for “rapidly fluctuating processes” like perception of pain, ability to cope with negative stimuli, and emotional affect [3]. Self-reported EMAs have been explored in the context of mobile health through apps like “TrackYourTinnitus” [4,5], where the users of the app are notified a selected number of times at random intervals during the day to answer a short questionnaire. In other cases, the answering of the EMA questionnaire may also be self-initiated. The data collected through such apps have already yielded interesting observations, like the effect of the time-of-day on tinnitus [6], the difference between users on the Android and iOS platforms [7], and the possibility of the emotional state of the patient mediating the influence of tinnitus loudness on the emotional distress [8].

This study uses EMA data generated by the “TinnitusTips” application (built on the TinnitusCare Framework), which combines the features of recording EMAs with a psychoeducation module. The psychoeducation module delivers a randomly selected ‘tip’ to the user upon the successful completion of the EMA questionnaire. Each tip has a goal, and a text on how to better achieve that goal (e.g., “Falling asleep better”) and helps the user to better handle their tinnitus. The data used in this work was generated as part of a user study meant to investigate the effect of the tips on a set of previously selected users who are admitted into the study. The set of users in the study are split into two groups, one of whom are exposed to the tips from the beginning of their interactions with the app, and the other who do not receive the tips for the first 8 weeks of app-interaction but subsequently receive the tips. The fact that some users are exposed to the tips after a fixed amount of interaction time makes it easy to isolate the effect of the tips themselves since the same user has been exposed to the app both with the ‘tip’ feedback (which can be seen as an intervention) and without it. We investigate the following questions in this study:To what extent is the known positive relationship between tinnitus loudness and distress reflected in the EMA data?How does the introduction of tips affect user behaviour within the system?How does the introduction of tips affect the user’s symptom severity?

Question 1 helps understands if EMAs ‘work’, in that the values they reflect known relationships between loudness and distress in Tinnitus. Questions 2 & 3 investigate if the introduction of tips cause some change in the patients, either in their behaviour within the app, or in the way they experience their symptoms.

## 2. Materials and Methods

### 2.1. The TinnitusCare mHealth framework

Figure 1 presents an overview of the TinnitusCare mHealth framework, which includes elements of psychoeducation for chronic tinnitus patients along with EMA items to help the patients monitor the fluctuations of their own tinnitus. Technically, the framework encompasses a relational database, a RESTful API, and an iOS mobile application. The latter was developed as a native implementation in the SWIFT programming language. The app is available in English and German.

All participants in the study were patients from the University Clinic at Regensburg, and were randomly selected into the three groups A, B and Y. All participants provided an informed consent regarding the use of the data they enter into the app. Before starting interaction with the app, study participants registered with the platform and filled registration questionnaires. Then, patients have to allow or decline GPS and environmental sound measurements. The *psychoeducation component* involves tips about dealing with tinnitus. All tips were structured the same way: First, there was the goal of the tip defined (e.g., better falling asleep), then there was a specific tip given on how to reach this goal (e.g., listening to music), and third, an explanation was given as to why the tip helps (e.g., music can mask the tinnitus).

### 2.2. The Questionnaires of the Intervention

The EMA questionnaires are designed to assess tinnitus and capture the user’s current context with several short questions during everyday life. The daily assessment questionnaires constitute the main part of this EMA study. The smartphone app will notify the user at several time points during the day to fill out the daily assessment questionnaires. The questionnaire comprises 8 questions, and is expected to take less than a minute to complete. The questions are shown in Table 1, along with the question codes that we refer to them with.

To assess the impact of the intervention, the study encompasses the following registration questionnaires, to be filled at the beginning of the study:Mini-TQ-12 [9]: tinnitus-related psychological problems (12 items)Tinnitus Sample Case History Questionnaire (TSCHQ) [10]: Current tinnitus status and demographic data (35 items)Worst symptom questionnaire: worst symptom (1 item)

The questionnaire is designed as a part of Ecological Momentary Assessment by handheld devices which has been used for studying persistent diseases with high variability in their symptoms and distress [11,12]. This daily questionnaire from TrackYourTinnitus (TYT) smartphone application has been analysed in previous studies [4,5,13]. Two different scales were used for these questions, i.e., a binary scale for questions 1 and 8 with ‘Yes’ and ‘No’ as possible answers, and visual analogue scale for Q2 to Q7, ranging from 0 to 100.

In addition to the data in the app, the patients also filled in the Tinnitus Handicap Inventory (THI) [14] and the WHO Quality of Life Questionnaires (WHOQOL-BREF) during their screening, mid-study, post-study and follow-up visits to the university clinic at Regensburg. Unlike the in-app questionnaires, there are filled in at the clinic. This helps track patient progress as measured by the EMAs against validated questionnaires as they pass through the study.

### 2.3. Dataset

The data generated by the study in question consists of users randomly selected into three groups: A, B, and Y. Users in groups A and Y were both exposed to tips from the beginning of the study, while users in group B did not initially receive tips, and began receiving tips only from their 8th week. Since groups A and Y were both exposed to tips since the beginning, and differ only in supervisor and number of assessments, we focus on comparisons between groups A and B since they are very similar in number of participants (A: *N* = 11 and B: *N* = 10) compared to group Y (*N* = 15). A tabular description of the main features of the dataset is presented below in Table 2.

The EMA dataset contained a further 1408 observations from 20 users who do not fall into any of the categories above. Since the additional variables regarding the patients are obtained by matching the user-id from the app with a hospital database, the age and gender of these users are not known. These users are excluded from the analysis, and all further references to the dataset refer to the dataset including only the users from groups A, B, and Y.

The patients in each of the groups had also filled in the THI questionnaire during their first assessment. The values at the baseline visit for the three groups are as shown in Figure 2.

### 2.4. Methods

The multiple answers from a participant to the EMA questionnaire creates an 8-dimensional time series. We refer to each individual response of the participant to the EMA questionnaire as a *session*. The main variables of interest in this study are listed below:Number of sessions contributed by a userValues of the answers to the questions s02 and s03 (tinnitus loudness and tinnitus distress respectively, see Table 1)

To address question Q2 from Section 1, we define variables *User Loyalty*, *User Engagement*, and *Dropout rate*. *User loyalty* is the total number of days on which the user logged at least one session, and *user engagement* is the total number of days on which the user contributed more than one session. The dropout rate is the number of users who remain active after a given number of weeks. The three variables model the *user behaviour* and are expected to reflect the effect of receiving tips.

However, before investigating the effect of the intervention on the participants’ subjective experience of tinnitus as measured by their EMAs, we first investigate the extent to which the EMA data reflects a known relationship between the relationship between tinnitus loudness and tinnitus distress [8].

#### 2.4.1. Q1: Do EMAs Reflect the Known Relationship between Tinnitus Loudness and Distress?

Although there is a known impact of tinnitus loudness on distress, there is evidence to suggest that there might be participants who have low values for distress in spite of relatively high values for tinnitus loudness [15]. To investigate the degree to which tinnitus distress is affected by loudness, we cluster the EMA responses to find groups of responses that are similar and then build a linear regressor with distress as the target for the data within each cluster. The parameters of the learned model can show the degree to which the distress depends on loudness. We also maintain a *hold-out set* of EMA responses, which is a randomly selected subset of all EMA responses to validate the model. This hold-out set has not been used in the clustering process or to train the cluster-level linear regression models.

The model parameters alone (even if they are significant) don’t give a compelling argument for building clusters where the relationship between loudness and distress needs to be modelled separately. As an additional step, we also train a single ‘global model’, trained on the data used to create the clusters, and test how well this global model predicts the data in the hold-out set. This error is compared against the errors from predicting the holdout data from each of the cluster-level models. If the cluster-level models show a better prediction accuracy than the global model, it can further strengthen the argument in favour of the clustering process and the model parameters learned by the cluster-level regression models.

In this work, we cluster the data using K-Means [16], a popular and mature algorithm for clustering. It works by randomly selecting *k* points from the input as *centres* of clusters, and assigns each of the rest of the points to the cluster centre nearest to it as measured by the euclidean distance between them. The cluster centre is then updated as the arithmetic average (geometrically, the ‘middle’) of all the points assigned to it. Now that the centre has shifted, points near the border of the cluster may now be closer to another cluster (causing a change in that cluster’s centre, and so on). This process is repeated iteratively for all points and clusters until the points assigned to a cluster do not change. The quality of the clustering process is assessed by computing the total squared distance of each of the points to the centre of the nearest cluster. To mitigate the effects of random initialisation on the final clustering (i.e., so that the whole clustering process is not handicapped by a poor selection of points at the initial step), the entire clustering process is repeated multiple times and only the best result is used. The exact value of *K* in K-Means is selected by a human in a way that balances the number of discovered clusters with the number of data points within them, and how well they reveal underlying groups in the data.

#### 2.4.2. Q2: Analysing User Behaviour

To compare the users from the two groups, we select two complementary measurements over which they can be compared: (a) User loyalty: Total number of days the user was active on the app (i.e., number of days on which the user has contributed at least one session), and (b) User engagement: The number of days on which the user has contributed *more than one* session. This measures interaction intensity.

The user loyalty for users in group B is compared to that of group A by using a box-and-whiskers plot of the number of days of data contributed by a user against the group they belong to. Each day on which a user has answered at least one EMA question is counted as a day on which the user was ‘active’. Since users belonging to each group have a specific ‘length’ to their time series (the number of days on which the user logged at least one session), the two distributions of time series lengths are compared using a Wilcoxon rank sum test to check if they are generated by the same underlying distribution.

The test for user engagement is done similarly, comparing the groups A & B on whether they differ significantly in the number of days on which they answered the EMA questionnaires more than once.

#### 2.4.3. Q3: Analysing Effect of Intervention on Tinnitus Symptoms

We investigate the effect of the tips on loudness and distress using a two-sample Kolmogorov-Smirnov test on the values for loudness before and after the introduction of tips, and similarly for distress. If the test reveals that the data before and after the introduction of tips are not generated by the same underlying distribution, it can be argued that the introduction of tips affects the expression of tinnitus for those patients.

## 3. Results

Figure 3 shows the data distribution of the answers to each questionnaire item for the users in each of the groups Y (*N* = 15), A (*N* = 11) and B (*N* = 10). We observe that group B is different from the other two groups, namely (i) the likelihood of S01 = NO is smaller for group B than for Y, A, and (ii) higher values of S02, of S03 and S04 are more likely in group B than in Y, A.

### 3.1. Q1: Does EMA Data Reflect the Impact of Loudness on Distress?

The clustering step is achieved through the use of the k-means algorithm. Unfortunately, the nature of the dataset is not conducive to the discovery of well-separated clusters, and the clustering here plays the role of a data-driven segmentation of the answer-space into cohesive groups. The results of the k-means clustering algorithm is shown in Figure 4. The number of clusters *k* was set to 5, in a way that balances interpretability while holding the number of clusters as low as possible.

When training separate regression models to study the relationship between loudness and distress in each of these separate clusters, the results are as shown in Table 3. For all clusters except Cluster 2, the model trained on the data from the cluster is significant.

The data selected for training the model is a subset of the total data available. This allows for evaluating not only on the model parameters but also to evaluate the model by testing its predictive power on data heretofore unseen during the training process. The applicability of the clustering step can also be confirmed by comparing the predictive power of a model trained on the entire dataset to the predictive performance of each of the separate models. Figure 5 shows the RMSE values for the predictions from each of the clusters

### 3.2. Q2: User Behaviour

A comparison of the groups A, B and Y based on the lengths of their time series are shown in Figure 6, where it can be seen that users in group B stay significantly longer in the system. A Wilcoxon rank sum test comparing groups A and B verify this difference with *p* = 0.037, further increasing to *p* = 0.008 if the one user in group A with more than 250 observations is excluded (this user is active for a total of 144 more days than the user with the next highest number). A comparison with the number of days for which the users log more than one session reveals a similar result, which is shown in Figure 7. The Wilcoxon rank sum test comparing groups A and B shows that the means are different with *p* = 0.0366.

The *dropout rate* for groups A, B and Y can be seen to be 4, 2 and 7 respectively. Group Y exhibited the highest dropout rate of $\frac{7}{15}$ participants, while group B exhibited the lowest, with only $\frac{1}{9}$. The *interaction intensity*, defined as the average number of sessions in each week, decreased for all three groups. All values are shown in Table 4.

In addition to studying the differences between users in groups A & B using only their EMA data, we also show the results between the users of each group as measured by the THI questionnaire collected during the baseline, mid-study, post-study and follow-up visits, as shown in Figure 8. The Mini-TQ questionnaire, unfortunately, has too many missing values to be included. Compared to the THI questionnaire which has almost no missing values, the poor adherence to filling in the in-app Mini-TQ questionnaire is likely because it is not filled in at the clinic, in the presence of a medical supervisor.

### 3.3. Q3: Effect of Intervention on Tinnitus Symptoms

To assess the impact of the introduction of tips, we perform a Two-Sample Kolmogorov-Smirnov test on the values of (a) tinnitus loudness before and after the introduction of tips, and (b) tinnitus distress before and after the introduction of tips.

Both tests reject the null hypothesis that the data from before and after the introduction of tips are generated by the same distribution (*p* = 9.9e-16 for loudness and *p* = 8.9e-12 for distress), indicating that the introduction of tips does change the EMAs observed for loudness and distress. Figure 9 shows the EMAs generated before and after the introduction of tips.

## 4. Discussion

The results in Section 3 ask three orthogonal questions—Do the EMAs reflect the known impact of tinnitus loudness on distress, whether the intervention affects the way in which the user interacts with the app, whether the intervention changes the user’s perception of the disease.

Regarding the impact of tinnitus loudness on distress, the diagonal band in Figure 4 confirms that loudness and distress are indeed highly correlated variables. However, the cluster of data points (shown in red) in the bottom right corner is especially interesting, since it shows that there are times when the respondents report low tinnitus distress in spite of relatively higher levels of tinnitus loudness. It can also be visually intuited that the data at the low-loudness, low-distress corners are less noisy than those in the middle of the band. $Model_{all}$ shows that a unit increase in loudness causes a 0.76 unit increase in distress, but the cluster level models $Model_{c0}\ldots Model_{c4}$ show that the distress-impact of a unit increase in loudness may vary from 0.12 to 0.61, depending on the exact value of the loudness.

It is also seen that observations from the different clusters are not equally easy to predict, i.e., the nature of the interaction between tinnitus loudness and distress is not always the same, and it depends on symptom severity. It is to be noted that even the models with relatively poor R-squared values (for example, the model on cluster 2 which does not show relationships strong enough to reject the null hypothesis) do show better predictive power over unseen data than the single model trained over the combined data from all clusters. This supports the suspicion that the strength of interactions between loudness and distress does depend on the absolute levels of symptom strength, and that while tinnitus loudness does impact distress, the strength of this interaction may not always be the same.

The results in Figure 6 show that users of group B who were exposed to the intervention were in fact persuaded to continue interacting with the app for longer than the users of group A. The fact that both groups A and Y are similar to each other and different to B supports the fact that it is indeed the intervention that causes the change in behaviour.

While the length of time for which a user engages with the system is an important metric, the user engagement is also relevant. Figure 7 suggests that not only are users of group B using the app for longer, but also that they are, on average, more active per day (they answer more EMAs). This is also visible over all the data, where the users in Group A log on average 1.79 sessions per day, and those in Group B log 2.40 sessions per day. In addition to changing the intensity of the user’s interaction with the app, the Kolmogorov-Smirnov test on the loudness and distress values before and after the introduction of tips shows that the expression of tinnitus symptoms for users of group B is also affected.

In addition to changing the user behaviour within the app, the introduction of tips is also seen to change the subjective experience of both tinnitus loudness and tinnitus distress. However, comparing this result to Figure 8 makes it difficult to draw concrete conclusions—though the participants of group B do show lower THI values mid-study, the benefits do not seem to persist later on into the study.

*Threats to validity:* The current study leaves open several avenues for improvement, the first of which is that there was no group of users who never received tips. To truly assess the impact of tips on the subjective experience of tinnitus, it would be necessary to recruit such a group of participants who use the app with the tips feature disabled. Making some in-app registration questionnaires like the Mini-TQ mandatory would also enable comparison of the app-based EMA data with validated questionnaires. Questionnaires like Mini-TQ can also be requested from the participants multiple times during the study to better assess whether changes to the EMA questionnaire answers are accompanied by clinically relevant changes. This becomes especially important when considering the fact that though we see a significant difference in the loudness and distress after the introduction of tips, the small mid-study improvement in THI-scores for group B reverts at post-study to values similar to baseline. The number of patients can also be increased, and since the participants are recruited at a hospital, adding features like audiological examinations and other tests may provide features that help us identify similarities in the EMAs depending on patient characteristics. Since Tinnitus is a disease that affects older people disproportionately, the use of an app in itself might be a factor that causes some people to drop out, causing a bias in the EMA data that is collected. Another useful addition to the analysis would be to assess the impact of other variables from the EMA on tinnitus distress, not just loudness.

To conclude, it is clear that users who started receiving tips to deal with their tinnitus do indeed stay active on the system for longer. However, since users of group A have also received the tips, it is possible that the introduction of the tips after the user is already acclimatised to the app makes it more beneficial to the user. Another possibility is that the increased user engagement is driven by the introduction of a new feature in the app, which sustains user interest. While this cannot be verified with the available data, it still does not explain why the introduction of tips is accompanied by a significant change in the subjective perception of both tinnitus loudness and distress. Visualisations of the EMA responses like that shown in Figure 9 further underscore the need for continued efforts in recording EMA data, since other questionnaires, especially those that are answered in a hospital setting, may not capture changes of this nature.

## Figures and Tables

**Figure 1 brainsci-10-00924-f001:**
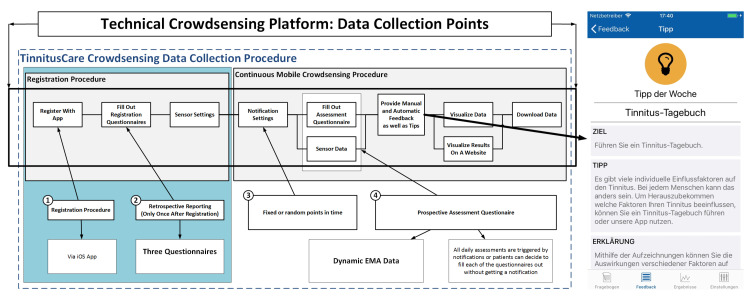
Mobile Crowdsensing Collection Procedure with a Psychoeducation Component.

**Figure 2 brainsci-10-00924-f002:**
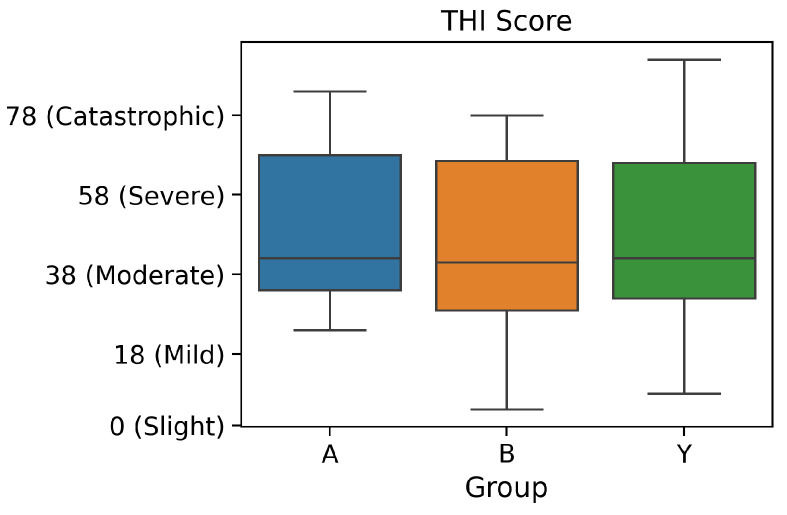
THI Scores at Baseline: Groups A, B and Y.

**Figure 3 brainsci-10-00924-f003:**
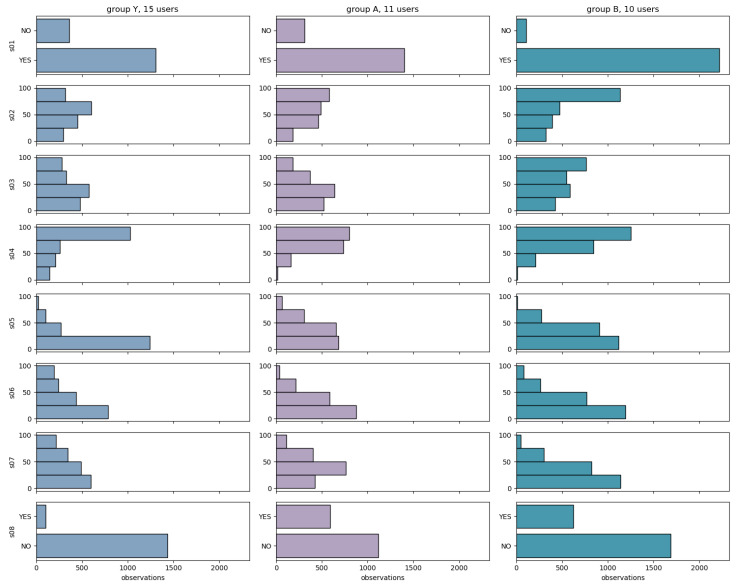
Data distribution of the answers to each questionnaire item across the groups Y, A and B.

**Figure 4 brainsci-10-00924-f004:**
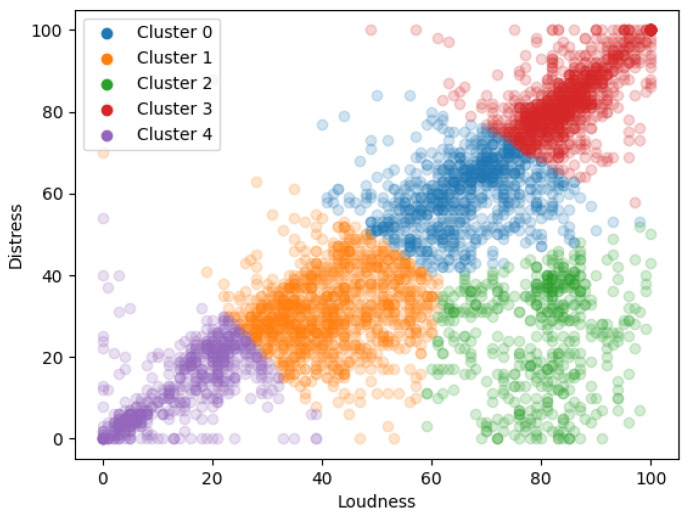
The 5 clusters of loudness/distress answers.

**Figure 5 brainsci-10-00924-f005:**
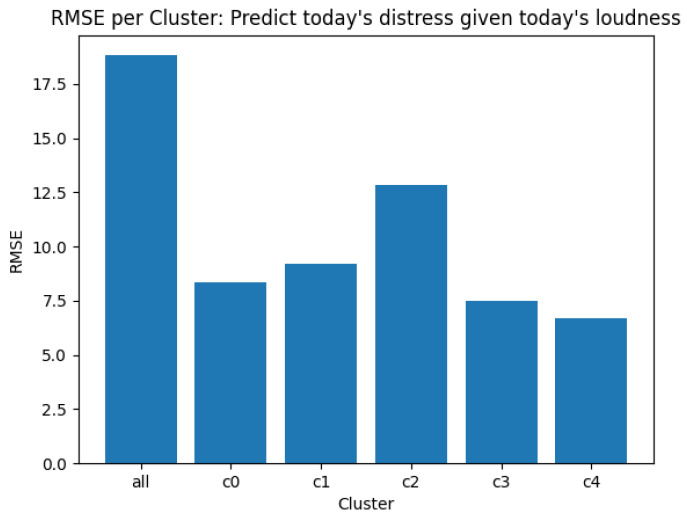
Out-of-sample prediction errors: Comparing the cluster-model errors against a model trained on all data.

**Figure 6 brainsci-10-00924-f006:**
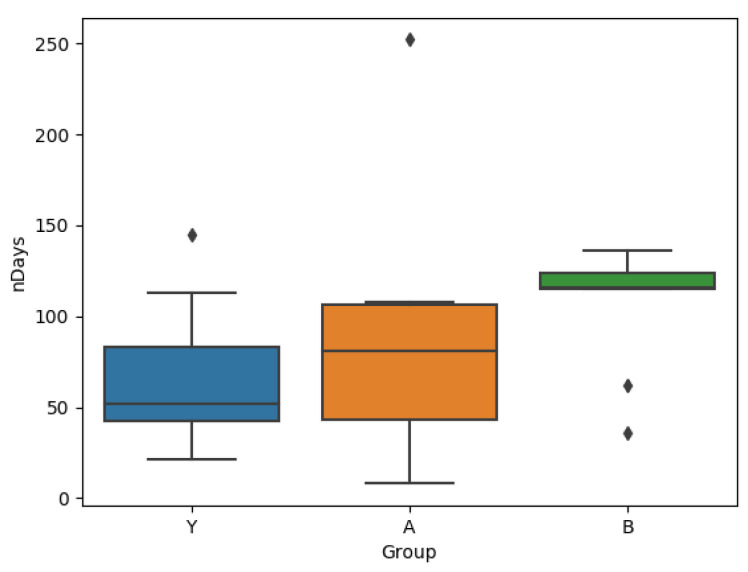
Box plots for time series length for each group.

**Figure 7 brainsci-10-00924-f007:**
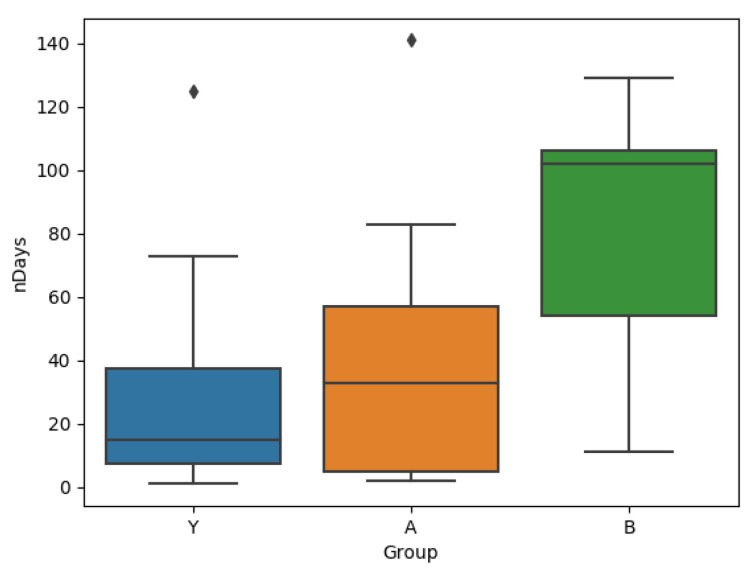
Box plots for number of days with 2 or more sessions for each group (Note: Y-axis not aligned with the previous plot).

**Figure 8 brainsci-10-00924-f008:**
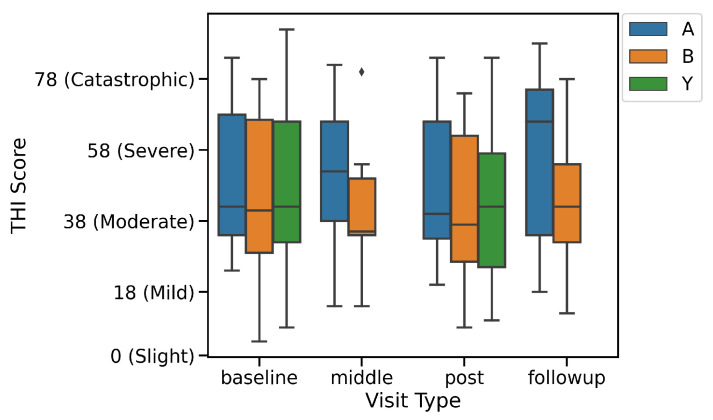
THI Score during baseline, mid-study, post-study and follow-up visits for groups A, B and Y.

**Figure 9 brainsci-10-00924-f009:**
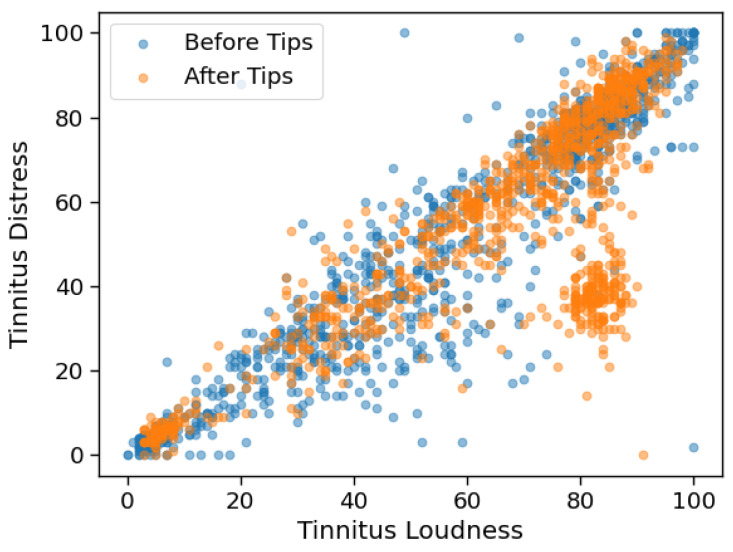
Plot showing loudness v/s distress EMAs before and after the introduction of tips.

**Table 1 brainsci-10-00924-t001:** EMA Questionnaire: Questions and scales for the answers to the EMA questionnaire.

Code	Question	Type
s01	Do you perceive the tinnitus right now?	Binary (Yes/No)
s02	How loud is your tinnitus right now?	Numeric (0–100)
s03	How distressed are you by your tinnitus right now?	Numeric (0–100)
s04	How well do you hear right now?	Numeric (0–100)
s05	How much are you limited by your hearing right now?	Numeric (0–100)
s06	How stressed do you feel right now?	Numeric (0–100)
s07	How exhausted do you feel right now?	Numeric (0–100)
s08	Are you wearing a hearing aid right now?	Binary (Yes/No)

**Table 2 brainsci-10-00924-t002:** User Groups A, B & Y: Basic Statistics.

	Group A	Group B	Group Y
Number of users	11	10	15
Age (± stdev)	49.63±9.84	51.70±12.72	47.6±12.65
Sex (m/f/*)	7m, 4f	6m, 4f	9m, 6f
Total EMAs recorded	1689	2286	1744

**Table 3 brainsci-10-00924-t003:** Model Parameters for each of the five clusters, compared to a model learned over all data without using clustering. All parameters are statistically significant with *p* < 0.001 except otherwise specified.

	Prob (F-Statistic)	R-Squared	Intercept	Loudness
$Model_{all}$	< 1 × 10^−101^	0.535	2.59	0.76
$Model_{c0}$	1.35 × 10^−9^	0.046	47.2	0.192
$Model_{c1}$	6.29 × 10^−5^	0.015	26.44	0.124
$Model_{c2}$	*0.0253*	0.01	*13.59 (p = 0.025)*	*0.147 (p = 0.010)*
$Model_{c3}$	1.11 × 10^−101^	0.385	23.34	0.407
$Model_{c4}$	1.25 × 10^−60^	0.367	4.46	0.61

**Table 4 brainsci-10-00924-t004:** Average number of sessions for each group for the participants still active in week 1, week 8 and week 16.

Group	Average Number of Sessions		
	Week 1	Week 8	Week 16
A	10.64 (*N* = 11)	7.2 (*N* = 10)	7.3 (*N* = 7)
B	19.56 (*N* = 10)	13.44 (*N* = 9)	12.0 (*N* = 8)
Y	14.53 (*N* = 15)	8.8 (*N* = 10)	5.6 (*N* = 8)

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
