# Peer review of "The Effect of Non-Personalised Tips on the Continued Use of Self-Monitoring mHealth Applications"

_brainsci, 2020, doi:10.3390/brainsci10120924_

Round 1

Reviewer 1 Report

This study highlight the important role of mobile health
for tinnitus patients especially during COVID-19 pandemic
but all sections must be improved to better understanding
the positive effects of mHealth applications.
Introduction:
-line 17, tinnire not tinniere
- tinnitus can also be related to sudden hearing loss, acoustic
trauma, noise exposure, ototoxic drugs, ecc.

Methods:
-In which languages the app is available?
- The informations about the sample are very poor: did you
select tinnitus patients from the hospital? did you include
patients with chronic tinnitus during the COVID-19 pandemic? did
you consider some exclusion criteria in the selection of the
patients? the supervisor of the groups was a psychologist?

Results:
- the results of the mini-TQ are not reported: how many tinnitus
patients were classified as decompensated based on mini-TQ score?.
- there is a lack of relevant clinical data from TSCHQ and mini-TQ
that could be presented in a table.

Discussion:
-limitations of the study: small sample size and no audiological data.
-could you underline the core of your results in comparison to
previous studies?

Reference n.4 and n.14 are the same.

Author Response

Thank you for the detailed review. It helped me identify some crucial issues with the submission... The review was very helpful in restructuring my work, especially as an outsider to the field

Reviewer 2 Report

An interesting Application that could be helpful in linking doctors to patients, so that we can monitor patient care more closely.

The paper is well presented, with excellent use of the English language. Minor revision is needed (e.g on line 217 group C is probably group Y)

Questions arise on whether the application will be easy to use amongst elderly patients who are more likely to suffer from tinnitus, however the dropout rate was significantly lower on the oldest average group. 

Finally, a larger number of patients should be included in the study so that the results could be statistically significant. 

Author Response

Thank you for your review. We found the review process very helpful in fixing some holes in our argument (especially for me, as computer scientist)

Reviewer 3 Report

The manuscript deals with a subject that has clear clinical relevance.

The weak points of the study are:

  • The English language needs corrections
  • The patients in each group are few
  • The Methods are appropriate, but they are not easy to be followed
  • The Results should be presented more clearly
  • The clinical relevance of the study should be explained more explicitly.

Author Response

Thank you for your review. We hope the revised version is easier to understand.

Round 2

Reviewer 1 Report

Thank you, the manuscript has been significanly improved. The idea to add the results of TSCHQ as an appendix is really appreciated.

Author Response

Thank you for your suggestions.

The TSCHQ information has been added as 'supplementary material' at the group-level so that data cannot be deanonymised. The 'date of birth' column ( TSCHQ Question 1) has been replaced with a manually computed 'age' (again, for purposes of preserving anonymity), some columns had textual answers that were impossible to tabulate at the group level.

Another reading pass and Grammarly-check has been done before submission.